# Floodborne Objects Type Recognition Using Computer Vision to Mitigate Blockage Originated Floods

Umair Iqbal [1,*], Muhammad Zain Bin Riaz [2], Johan Barthelemy [3], Nathanael Hutchison [1] and Pascal Perez [1]

1 SMART Infrastructure Facility, University of Wollongong, Wollongong, NSW 2522, Australia
2 School of Civil, Mining and Environmental Engineering, University of Wollongong, Wollongong, NSW 2522, Australia
3 NVIDIA Incorporation Ltd., Santa Clara, CA 95051, USA
* Correspondence: umair@uow.edu.au

**Abstract:** The presence of floodborne objects (i.e., vegetation, urban objects) during floods is considered a very critical factor because of their non-linear complex hydrodynamics and impacts on flooding outcomes (e.g., diversion of flows, damage to structures, downstream scouring, failure of structures). Conventional flood models are unable to incorporate the impact of floodborne objects mainly because of the highly complex hydrodynamics and non-linear nature associated with their kinematics and accumulation. Vegetation (i.e., logs, branches, shrubs, entangled grass) and urban objects (i.e., vehicles, bins, shopping carts, building waste materials) offer significant materialistic, hydrodynamic and characterization differences which impact flooding outcomes differently. Therefore, recognition of the types of floodborne objects is considered a key aspect in the process of assessing their impact on flooding. The identification of floodborne object types is performed manually by the flood management officials, and there exists no automated solution in this regard. This paper proposes the use of computer vision technologies for automated floodborne objects type identification from a vision sensor. The proposed approach is to use computer vision object detection (i.e., Faster R-CNN, YOLOv4) models to detect a floodborne object's type from a given image. The dataset used for this research is referred to as the "Floodborne Objects Recognition Dataset (FORD)" and includes real images of floodborne objects blocking the hydraulic structures extracted from Wollongong City Council (WCC) records and simulated images of scaled floodborne objects blocking the culverts collected from hydraulics laboratory experiments. From the results, the Faster R-CNN model with MobileNet backbone was able to achieve the best Mean Average Precision (mAP) of 84% over the test dataset. To demonstrate the practical use of the proposed approach, two potential use cases for the proposed floodborne object type recognition are reported. Overall, the performance of the implemented computer vision models indicated that such models have the potential to be used for automated identification of floodborne object types.

**Keywords:** blockage of hydraulic structures; computer vision; object detection; floodborne objects; floods

## 1. Introduction

Floods are one of the natural disasters which usually occur on a large scale and result in catastrophic damage to the community [1,2]. Rapid evacuations, damage to property, wildlife loss, human causalities and agricultural damage are a few of the most highlighted damages from a flooding event [3,4]. The frequency of rain-originated floods has been observed to be increasing over the last couple of decades mainly because of an increase in the duration and intensity of rainfalls and blockage of urban drainage structures. The rain is a naturally occurring phenomenon and hence cannot be controlled; however, drainage structures (e.g., bridges, culverts, sewerage) can be efficiently managed to avoid urban floods to a significant extent. Hydraulic structures including bridges and culverts are vulnerable to

damage and ultimately fail during extreme floods. A few highlighted reasons for the failure of hydraulic structures include huge hydrodynamic loads during transient flows [5,6], downstream/piers scouring by extreme flows [7–9] and interactions of floodborne objects. These factors of hydrodynamic loads, scouring and fllodborne objects individually or collectively result in sustained damage to hydraulic structures. In the context of urban flash floods, blockage of drainage structures by the floodborne objects are found to be one of the key factors which originate floods [10–13]. The Wollongong, Australia, 1998 floods [13,14]; Newcastle, Australia, 2008 floods [13,15]; and Pentre, United Kingdom, floods [16] are examples from the recent past where floods resulted because of hydraulic structures were blocked or failed by the accumulation and interaction of floodborne objects. The interaction of floodborne objects with hydraulic structures during extreme flows can impact the flooding outcome in many ways. During extreme floods, floodborne objects can cause large impulsive loads on the bridges [17] and simple structures [18]. Furthermore, the accumulation of objects across bridges and culverts often results in increased flow velocities and increased applied moments (e.g., pitch, yaw, roll), enhancing the likelihood of a structure to fail [19].

　　Conventionally, flood modelling through hydraulic engineering is an established science; however, it is often observed failing in practice, resulting in damages [20–23]. One of the reasons for the failure of conventional flood models is their inability to account for the impact of non-linear factors such as floodborne objects [24–29]. However, multi-physics numerical approaches (e.g., coupled SPH-FEM, coupled SPH-DEM) have recently demonstrated their potential for simulating debris effects. Hasanpour et al. [30] reported the use of the SPH-FEM multi-physics approach for the successful simulation of the impacts of debris movement and loads during floods. Furthermore, Trujillo-Vela et al. [31] also used a similar multi-physics approach (i.e., SPH-DEM) to simulate the debris in soil flows, which indicates the potential for the use of these approaches for floodborne debris simulation. Although multi-physics simulation approaches have shown encouraging performance, they are limited by the interaction and handling of multiple numerical methods for a single physical simulation. Furthermore, these models are dependent on the established fundamental equations defining the physics of the involved processes [32,33]. These models may be successful when dealing with a single type of floodborne object (e.g., tree logs, vehicle); however, in practice, the uncertain nature of the accumulation, combination and interaction of floodborne objects with hydraulic structures offer very complex hydrodynamic impacts. For such complex, non-linear and uncertain cases, as in floodborne objects, where there exist no apparent relationships, data-driven models have shown significant success in capturing the hidden relations among variables. However, given the limited availability of data related to floodborne objects and blockage of structures, the research in this domain is hindered to some extent. In literature, floodborne objects are often categorized as (a) vegetation, consisting of tree branches, tree logs and entangled grass, and (b) urban objects, consisting of vehicles, bins, shopping carts and building waste materials [13,34]. Both types of floodborne objects have a very distinct impact on floods from a flood management perspective, and hence it is very important to recognize the types of floodborne objects. The hydraulic impacts of vegetation largely depend on the type, size and accumulation pattern. For example, a tree with rootwads offers more porous characteristics in comparison to the tree without rootwads; therefore, large tree logs without roots are often considered critical [35]. Furthermore, the diameter and length of the vegetation relative to the opening/size of the hydraulic structure also plays a vital role in and influences the depth of the flow required to entrain/transport the vegetation [36,37]. The density and transport regime of the vegetation are also important factors in the context of the blockage and flooding [37–40]. Generally, the smaller type of vegetation including grass, leaves and branches type of is not considered hydraulically critical unless entrained with large wood or urban objects [10]. On the other hand, compact urban objects, including vehicles, bins and building materials, are often considered hydraulically critical because of their ability to instantly block or damage a structure on impact. Shopping carts are

porous in nature and often offer very low hydraulic blockage [10]; however, they serve as a means for capturing small vegetation. Therefore, it is very important aspect within the flood management domain to recognize the type of floodborne objects to incorporate it in the flood-related decision-making process. So far, in the literature, there is no automated approach reported for the recognition of the types of floodborne objects.

The research reported in this paper is motivated from the recent success of Artificial Intelligence (AI) approaches in multiple application domains within water resources (e.g., water depth measurement [41], water quality prediction [42,43], structural damage assessment [44,45], blockage detection [10–12], marine debris detection [46,47]). This paper proposes the use of computer vision object detection models for the recognition of floodborne object types from images to help mitigate blockage-related floods. In tge literature, the Faster R-CNN and You Only Look Once version 4 (YOLOv4) models have been identified among the most robust and fast object detection models based on deep Convolutional Neural Network (CNN) architecture [46–50]. The Faster R-CNN model is developed based on the idea of using the shared CNN features for Region Proposal Network (RPN) from the feature extraction layers, making it a computationally efficient model. Overall, Faster R-CNN is a two-stage unified model, where the RPN model proposes the regions while the Fast R-CNN model detects the objects using proposed regions [51]. On the other hand, the YOLOv4 [52] belongs to the class of single stage detectors and is the enhancement of the YOLOv3 [53] model with many new universal features, enhanced data augmentation techniques and genetic models for hyperparamter optimization. The YOLO series of models is designed to achieve a robust performance, specifically for mobile platforms, and is one of the first choices for hardware deployment using edge computers. In this context, multiple variants of the Faster R-CNN and YOLOv4 models have been trained using the Floodborne Objects Recognition Dataset (FORD) established from local council records and simulated hydraulics laboratory experiments. The performance of the models was evaluated using standard Average Precision (AP) and Mean Average Precision (mAP) measures. AP is computed as the model's precision weighted mean at each confidence threshold, while weight is the increase in recall from the previous threshold. The precision of the model is calculated using the Intersection of Union (IoU) threshold within the object detection models. mAP is used for the case where multiple class prediction is involved and computes the mean of all the classes' AP to provide a cumulative performance measure. The mathematical expression for mAP is given as follows in Equation (1):

$$mAP = \frac{1}{N} \sum_{i}^{N} AP_i \tag{1}$$

As part of the big picture, the intuition behind the presented research is to make use of the latest computer vision models towards extracting useful information about floodborne objects during floods and incorporate that information within the existing models to adapt to the impacts of floodborne objects. Furthermore, the scope of the research also extends to the maintenance of the blocked hydraulic structures during floods in which the types of floodborne objects accumulated around structures is critical information used for decision making. In this context, as a first step, in this paper, we proposed the automated recognition of the types of floodborne objects using computer vision object detection models. Therefore, the anticipated contribution of the presented research is to establish a floodborne objects recognition dataset for bounding box detection and the implementation of the latest computer vision object detection models for the first time towards recognizing the types of floodborne objects from images.

The rest of the article is organized as follows. Section 2 reports the review of the most related literature on the automated detection of floodborne objects. Section 3 presents the details about the materials and methods. First, the details about the collection and development of the floodborne objects dataset (i.e., FORD) are provided. Second, the background theoretical information about the computer vision object detection models used in this

research is presented. Finally, a detailed description of the adopted research approach is provided for the development of an automated floodborne object recognition system. Section 4 presents the experimental protocols and performance evaluation measures. Section 5 presents the quantitative and graphical results of the implemented computer vision models for floodborne object type recognition. Section 6 provides the discussions and insights into the experimental investigation including research implications, research limitations and potential future research directions. Finally, Section 7 summarizes the study by highlighting key outcomes from the experiments and listing potential future research directions.

## 2. Related Work

In this section, benchmark literature related to the detection of water floating debris using computer vision models is summarized. The literature review is structured in chronological order to highlight the advancements over time.

MacVicar and Piegay [54], in the year 2012, proposed the use of video camera monitoring system for efficient detection of wood passage and transport rates in river. A semi-manual video monitoring approach was adapted where conventional image processing techniques were used to extract the velocity and size of the wood for budgeting. The results clearly suggested a highly non-linear relationship. An increase in wood transport was reported with an increase in the discharge. Although promising results were reported related to video-based wood budgeting, the use of conventional approaches suggests its invalidity as a generalized solution. This approach may work specifically for a certain site, however, may drastically fail when applied to a new site. Benacchio et al. [55], in the year 2017, proposed the use of ground-camera-based image processing to monitor the wood delivery in rivers. The idea of automatically detecting the wood area and translating it into weight and flux was used. A conventional approach of using a classical machine learning Support Vector Machine (SVM) model for pixel-based classification was adopted to classify if a given grid cell within the captured image belongs to the wood. Although the study reported over 90% accuracy in predicting the wood, the use of conventional models suggest that this cannot be considered a generalized performance.

Lieshout et al. [48], in the year 2020, proposed a camera-based setup equipped with deep learning object detection models to detect floating plastic debris in rivers. The dataset used to train the deep learning models was collected from five different locations and was annotated for the floating plastic objects boxes (e.g., plastic bottles, plastic bags). Overall, the dataset (i.e., river image dataset, floating plastic dataset) consisted of around 1300 images with approximately 14,500 bbox annotations. The Faster R-CNN model was used for the detection of floating plastic debris and reportedly achieved an mAP of 68.7%. From the experimental investigations, it can be observed that the model was able to achieve reasonable accuracy, indicating the challenging nature of the dataset mainly due to the presence of small targets and water reflections. There was no comparison with previous literature reported to demonstrate the status of the presented research. Furthermore, no comparative experimental investigations were performed using multiple CNN models to justify the selection of the best one for the detection of floating plastic debris. In the context of debris type recognition, the camera setup and floating nature of debris are common factors; however, the types of debris being detected are not exactly the ones categorized under floodborne objects.

Cheng et al. [46], in the year 2021, proposed a comprehensive marine floating debris detection dataset called FloW and reported the performance of state-of-the-art object detection models for the proposed dataset. The dataset included 2000 images with 5271 floating debris bbox annotations. The latest object detection models, including DSSD, RetinaNet, YOLOv3, Faster R-CNN and Cascade R-CNN, were implemented. The Cascade R-CNN model was able to achieve the highest mAP of 0.43 with 3.9 FPS, while the DSSD model was the fastest, with an FPS of 28.6 and an mAP of 0.275. From the experimental investigation, it can be observed that even the latest object detection models were found struggling and achieved relatively lower detection accuracy for the challenging dataset. The highlighted

challenges of the dataset include the presence of small objects in the water, water reflections, and reflections from other objects. In the context of floodborne objects detection, although the debris type being detected is not directly related, the water environment and floating nature of debris are common factors in both cases.

Ghaffarian et al. [56], in the year 2021, proposed the use of multiple image processing and tracking approaches for the automated detection and quantification of wood in a river. A combination of static masks, dynamic masks and tracking was used for the detection of wood in the used images. To further improve the false detection rate, post-processing techniques including precision improvement and estimating missed detections based on recall rate were used. From the results, it was reported that post-processing approaches were able to reduce the error rate to 15% from 36%, while the missed detection rate was reduced to only 6.5% from 71%. The conventional image processing approaches were used, and the potential of state-of-the-art deep-learning-based models was not explored. Furthermore, no comparison with the literature was made to compare the performance of the proposed methodology. In the context of floodborne object type detection, the problem of detecting wood in the river is very similar to what has been proposed in this manuscript about detecting floodborne vegetation.

Lin et al. [57], in the year 2021, developed an improved version of the YOLOv5 model with a Feature Map Attention (MAP) layer for the detection of floating debris to assess the water quality. The dataset used to train the model included 2400 images with floating objects from eight different classes (i.e., plastic bags, leaf, grass, bottles, milk boxes, balls, branches, plastic garbage). The dataset was further enhanced using a fusion approach where water backgrounds were merged with the floating debris objects. From the experimental investigations, the proposed model was able to achieve an mAP of 77% when compared with other models for the challenging dataset. The branch class in the dataset is most relevant to the floodborne vegetation type and was reported to be detected with an accuracy of 55% (i.e., lowest among all the eight classes), which indicates the challenges in detecting floodborne vegetation.

Most recently, Majchrowska et al. [58] proposed a detection–classification pipeline for the detection of waste material in natural and urban images. The efficientDet-D2 object detection model was used to detect the waste objects in the image, while the EfficientNet-B2 model was used to classify the detected waste objects into one of the eight waste classes. For waste detection, the benchmark detect-waste dataset was used along with a subset of many other similar datasets, including Extended TACO [59], TrashCan [60], Trash-ICRA [61], UAVVaste [62] and drinking-waste [63]. For the training of the classifier, a semi-supervised approach was adopted using thousands of images from the Open Litter Map [64] dataset. From the experiments, an mAP of 70% was reported for the waste detection using EfficientDet-D2, while an accuracy of 75% was reported for the classification of detected objects using EfficientNet-B2. Although acceptable accuracy was achieved from the results, there was no comparison of different detection and classification models reported to justify the selection of the implemented models. Furthermore, no comparison with literature was made to demonstrate the status of the reported research. The dataset used in the research is comprehensive; however, it is relatively easier, specifically designed for detection purposes with the presence of only the waste objects in the image, avoiding any background noise. In the context of floodborne object type recognition, this research is not directly related since it only attempts to detect the trash and small urban waste (e.g., plastic bottles, glass, accumulated trash, plastic bags) and not vegetation and large urban objects.

Aleem et al. [47], in their most recent research, proposed the use of a deep learning approach to detect marine debris. The Forward-Looking Sonar Image (FLS) marine debris dataset was used to train the computer vision object detection models. The dataset consisted of 1865 images belonging to 10 classes of marine debris. A Faster R-CNN model with ResNet50 and VGG16 backbones was used as an object detector to detect marine debris from images. From the experimental investigations, the Faster R-CNN model with ResNet50 was able to achieve a mean IoU of 3.78. There was no comparison reported with literature

to demonstrate the status of the presented research. In the context of floodborne object detection, the presented research is not directly relevant, but the water background in the marine images overlaps with one of the use cases (river use case) presented in this research.

In summary, most of the related literature addresses either the marine debris detection or the floating plastic debris detection problems, similar to the floodborne object detection problem. Faster R-CNN, DSDD, RetinaNet, YOLOv3, Cascade R-CNN, YOLOv5 and EfficientDet were the computer vision models reported for the detection of debris. The most relevant research in terms of detecting a similar type of vegetation was reported by Ghaffarian et al. [56] and Lin et al. [57]. Table 1 summarizes the benchmark literature related to the detection of different debris types using computer vision techniques.

**Table 1.** Summary of Benchmark Literature Related to the Detection of Different Debris Types using Computer Vision Techniques.

| Author | Year | Addressed Problem | Dataset | Proposed Approach | Performance |
|---|---|---|---|---|---|
| MacVicar and Piegay [54] | 2012 | wood detection | custom dataset | conventional methods | NA |
| Benacchio et al. [55] | 2017 | wood detection | custom dataset | conventional methods | $R^2$ of 93% |
| Lieshout et al. [48] | 2020 | floating plastic debris detection | custom dataset (1300 images) | Faster R-CNN | mAp of 68.7% |
| Cheng et al. [46] | 2021 | marine debris detection | custom dataset (2000 images) | DSSD, RetinaNet, YOLOv3, Faster R-CNN Cascade R-CNN | mAP of 43% for Cascade |
| Ghaffarian et al. [56] | 2021 | river wood detection | NA | Conventional static and dynamic masking | 21% improved error rate |
| Lin et al. [57] | 2021 | floating debris detection | custom dataset (2400 images) | Improved YOLOv5 | mAP of 77% |
| Majchrowska et al. [58] | 2022 | waste material detection | TACO, TrashCan, Trash-ICRA, UAVVaste, drinking-waste | EfficientDet-D2 EfficientNet-B2 | mAP of 70% |
| Aleem et al. [47] | 2022 | marine debris detection | FLS dataset (1865 images) | Faster R-CNN | IoU of 3.78 |

## 3. Materials and Methods

### 3.1. Floodborne Objects Recognition Dataset (FORD)

The dataset used in this research is referred to as the "Floodborne Objects Recognition Dataset (FORD)" and is developed mainly from two sources: (a) historical records of Wollongong City Council (WCC) and (b) hydraulic-lab-simulated experiments. Images in the dataset are the subset of the two datasets developed for visual blockage prediction: Images of Culvert Opening and Blockage (ICOB) [11] and the Visual Hydraulics-Lab Dataset (VHD) [11]. The ICOB dataset consists of the images of real culverts blocked with different types of floodborne objects. On the other hand, VHD includes the images from a detailed hydraulic study where multiple blockage scenarios were replicated using scaled physical models (see Iqbal et al. [10,12] for more details about the laboratory experimental setup). Given the fact that there exists no benchmark dataset related to floodborne object recognition, FORD is proposed to be first of its kind in this domain. However, the real images sorted from the WCC's historical records were very few and not enough to train the data-hungry deep learning models; therefore, the dataset was enhanced with simulated images captured from the laboratory experiments to facilitate the training process. Although the addition of controlled simulated dataset will have its own limitations, the investigation on the impacts of simulated data on the models performance is not in the scope of the presented research. For this specific study of recognizing the floodborne object types, the subset images from the ICOB and VHD were annotated for different floodborne object types and organized into a new dataset (i.e., FORD). There are, in total, 663 images (141 real, 522 simulated) in

the FORD with 946 bbox annotations (640 vegetated, 306 urban). Figures 1 and 2 show the annotated real and simulated sample images from FORD, respectively.

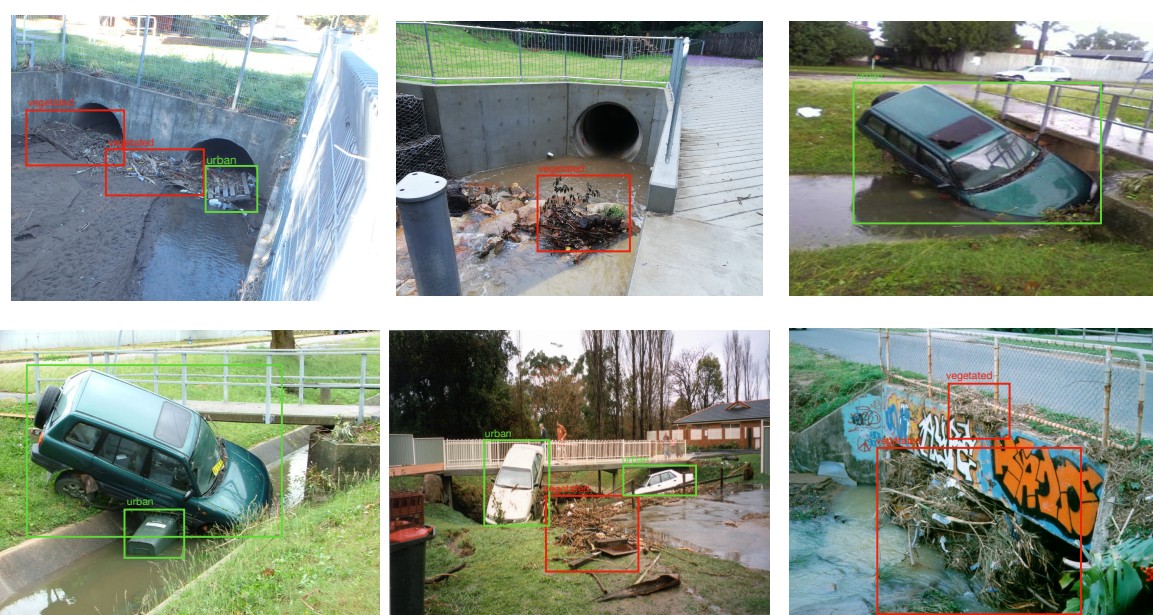

**Figure 1.** Annotated Real Samples from FORD.

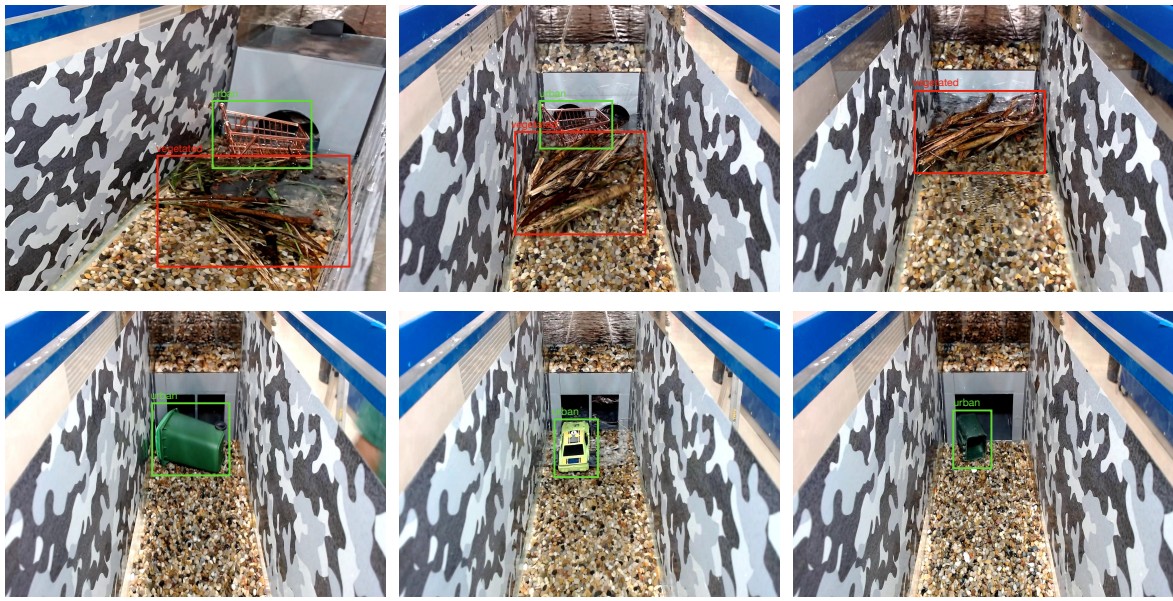

**Figure 2.** Annotated Simulated Samples from FORD.

### 3.2. Background to Computer Vision Object Detection Models

This section presents the theoretical background of the implemented computer vision object detection models (i.e., Faster R-CNN, YOLOv4) for floodborne object type recognition. Fundamental working principles and a description of the architecture for both modes are presented briefly.

#### 3.2.1. Faster R-CNN

Faster R-CNN is one of the most popular and commonly used object detection models proposed by Ren et al. [51] in 2017 to address the problem of region proposal computational

cost by proposing an RPN. The proposed RPN resulted in generating region proposals with almost no computational cost because of the idea of using the shared CNN features from the image to the detection network. Furthermore, the Faster R-CNN model merges the RPN and Fast R-CNN by sharing the convolutional features and using the attention mechanism. Overall, the Faster R-CNN model is a unified network with two modules: (a) a deep CNN architecture to propose the regions (RPN) and (b) a Fast R-CNN detector to use the proposed regions for detection. The RPN network generates the multi-scale anchors as regression references and built a pyramid type approach, which makes it more cost-efficient in comparison to other models. A multi-task loss function, as expressed in Equation (2) (i.e., a combination of classification and bbox regression loss), is used to optimize the training in the Faster R-CNN model:

$$\mathcal{L}(\{p_i\}, \{t_i\}) = \frac{1}{N_{\text{cls}}} \sum_i \mathcal{L}_{\text{cls}}(p_i, p_i^*) + \lambda \frac{1}{N_{\text{bbox}}} \sum_i p_i^* \mathcal{L}_{\text{bbox}}(t_i, t_i^*) \tag{2}$$

where $i$ represents the anchor index, $p_i$ represents the $i$th anchor probability, $p_i^*$ represents the $i$th anchor ground truth, $t_i$ represents the predicted bbox coordinates vector, $t_i^*$ represents the ground truth bbox coordinates vector, $N_{\text{cls}}$ and $N_{\text{bbox}}$ represent regularization terms and $\lambda$ represents the balancing parameter.

3.2.2. You Only Look Once version 4 (YOLOv4)

The YOLOv4 object detection model was proposed by Bochkovskiy et al. [52] in 2020 towards achieving optimized accuracy and speed by making use of multiple universal features including Cross-Stage Partial Connections (CSP), Self-Adversarial Training (SAT), mesh activation, Weighted Residual Connections (WRC) and Cross Mini-Batch Normalizations (CmBN). The YOLOv4 architecture was designed by selecting the optimal backbone network, neck network and head network for detection. The base YOLOv4 architecture included the CSPDakrNet53 backbone model, SSP additional module, PANet neck model and YOLOV3 head architecture for anchor-free detection. DropBlock has been used as the regularization method for the training for the YOLOv4 model. Some highlighted training-related improvements included the introduction of the new mosaic and SAT data augmentations and the selection of optimal hyperparameters using a genetic algorithm. As a result of these improvements, the YOLOv4 model was able to achieve improved performance over the benchmark dataset in comparison to its predecessor YOLOV3 while keeping the real-time functionality.

*3.3. Research Approach*

A five-step approach is adopted in this research to develop the computer-vision-based floodborne object type recognition solution (see Figure 3). A detailed description of the tasks and activities performed under each stage is provided as follows:

- Stage I: Data Preparation—At the first stage, the raw images from the WCC records were processed and annotated for training the computer vision object detection models. In context to data processing stage, firstly, the images from the records were manually sorted to select the suitable candidates for training. Presence of floodborne objects accumulated at culverts or within the catchment was used as the criterion to sort the images. Secondly, the selected images were cropped where required to remove the background noise and were converted to unified format for consistency. Once the final set of images was decided, they were annotated/labelled with ground truth bounding boxes of vegetation and urban objects in the images. For the labelling of images, an open source image annotation tool called LabelImg [65] was used, which, by default, saved the labels in XML format (i.e., one of the formats to which bounding box labels can be saved). Within the computer vision domain, there are different platforms developed to facilitate the training of the state-of-the-art models, including Detectron2, TensorFlow Object Detection API, NVIDIA Train Adapt Optimize (TAO) and DarkNet. Each of these platforms requires the ground truth labels to be stored in a specific

data format, for example, Detectron2 accepts .json format labels, TensorFlow API accepts XML format labels, NVIDIA TAO accepts KITTI labels and DarkNet accepts .txt format labels. For this research, NVIDIA TAO toolkit was used for training, which is a framework designed to simplify and accelerate the development of AI-oriented industrial solutions.

- Stage II: AI Development—At the second stage, the AI models were developed and trained using the labelled data from Stage I. In the process of AI development, firstly, the object detection models were selected from the available model zoo based on the performance reported in the literature. As a result, keeping the robustness and hardware deployment as key factors, Faster R-CNN (i.e., robust detection performance) and YOLOv4 (i.e., suitable for hardware deployment) model variants were selected to be trained for the floodborne object type recognition problem. Secondly, for each selected model, hyperparameters, including training epochs, learning rate, optimization function and regularization technique, were set using default off-the-shelf values. Furthermore, different data augmentation techniques (i.e., one of the conventional approaches used in computer vision model training where input image is subjected to different transformations towards creating multiple variants of same image) were also used during the training process to enhance the performance. All the models reported in this study were trained using the NVIDIA TAO platform.
- Stage III: Training Evaluation—At the third stage, the models were evaluated for their performance during the training phase using different standard evaluation measures including training loss per epoch, training time per epoch and validation mAP. In context of the deep learning computer vision models, the loss of a model refers to the prediction error (i.e., predicted-actual) and is a measure to assess how well a model has performed. In the training process, deep learning models use optimization functions (e.g., Stochastic Gradient Descent (SGD), Adaptive Momentum (adam)) with the objective to minimize the loss function using the backpropagation approach. The aim of assessing the training performance is to ensure that the training process did not involve any abnormal behaviour, specifically overfitting. Training loss and mAP curves are standard indicators to observe any abnormalities. Usually, for a normal training process, the loss curve should follow the negative exponential trend, while the mAP should follow the positive exponential trend.
- Stage IV: Test Evaluation—At the fourth stage, the trained object detection models were evaluated against the unseen validation data and were compared to identify the best performing model(s). Evaluation was performed using test mAP and AP for each of the two floodborne object classes.
- Stage V: Discussion—At the fifth and final stage, the inference results from the models, specifically with best test performance, were analysed and discussed in detail to report the important insights from the experiments. Furthermore, performance of the proposed approach was linked with existing literature, and different implications of the research were presented. In addition, potential limitations of the research were highlighted, and future directions were discussed.

## 4. Experimental Protocols and Evaluation Measures

The computer vision models were trained using a Linux machine having an NVIDIA A100 Graphical Processing Unit (GPU) with 80 GB memory. Python programming with TensorFlow and Keras packages was used for training the models. A dataset split of 80:20 was used for training and validation purposes, respectively. Two variants of the Faster R-CNN model (i.e., Faster R-CNN with ResNet18 backbone, Faster R-CNN with MobileNet backbone) and two variants of YOLOv4 (i.e., YOLOv4 with ResNet18 backbone, YOLOv4 with CSPDarkNet backbone) were trained for detecting vegetation and urban objects within the images. All the models were trained for 100 epochs with a batch size of 1. For Faster R-CNN models, data augmentation of horizontal flip, zoom variation and contrast variation were used. The models were trained using an SGD optimizer with a momentum of 0.9 and

a learning rate of 0.02. The L2 regularization approach was used for Faster R-CNN models. For the YOLOv4 models, data augmentation of colour transformation, horizontal flip and jitter were used. Adam optimizer with a learning rate of $1 \times 10^{-7}$ was used along with the L1 regularization approach. The training performance of models was assessed based on the training loss, validation mAP and training time in seconds per epoch. Moreover, the test performance of models was assessed using mAP and AP for each class.

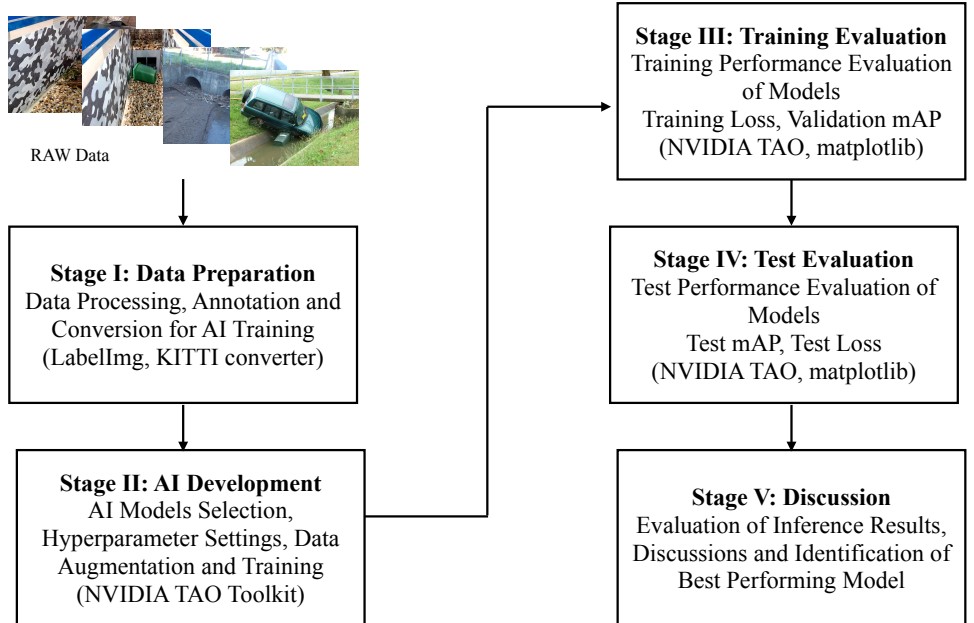

**Figure 3.** Functional Block Diagram of the Proposed Research Methodology for Floodborne Object Type Recognition.

## 5. Results

This section presents the quantitative and graphical results for the computer vision models to detect the floodborne object types. Computer vision models are assessed for their performance in both the training and testing phases.

### 5.1. Training Performance

The training performance of the models has been evaluated from the training loss curves, validation mAP curves, quantitative values from the best validation epoch and the training time taken by each model for a single epoch. The training loss and validation mAP curves for the Faster R-CNN models and YOLOv4 models are shown in Figures 4 and 5, respectively. For the Faster R-CNN models (see Figures 4a and 5a), it can be observed that both the ResNet18 and MobileNet variants performed comparatively, with the MobileNet variant being slightly on the better end. The loss curves followed the standard negative exponential trend and settled around 0.1, which is the indication of normal training performance. The validation mAP, however, stabilized to some extent after 40 epochs, around 80%.

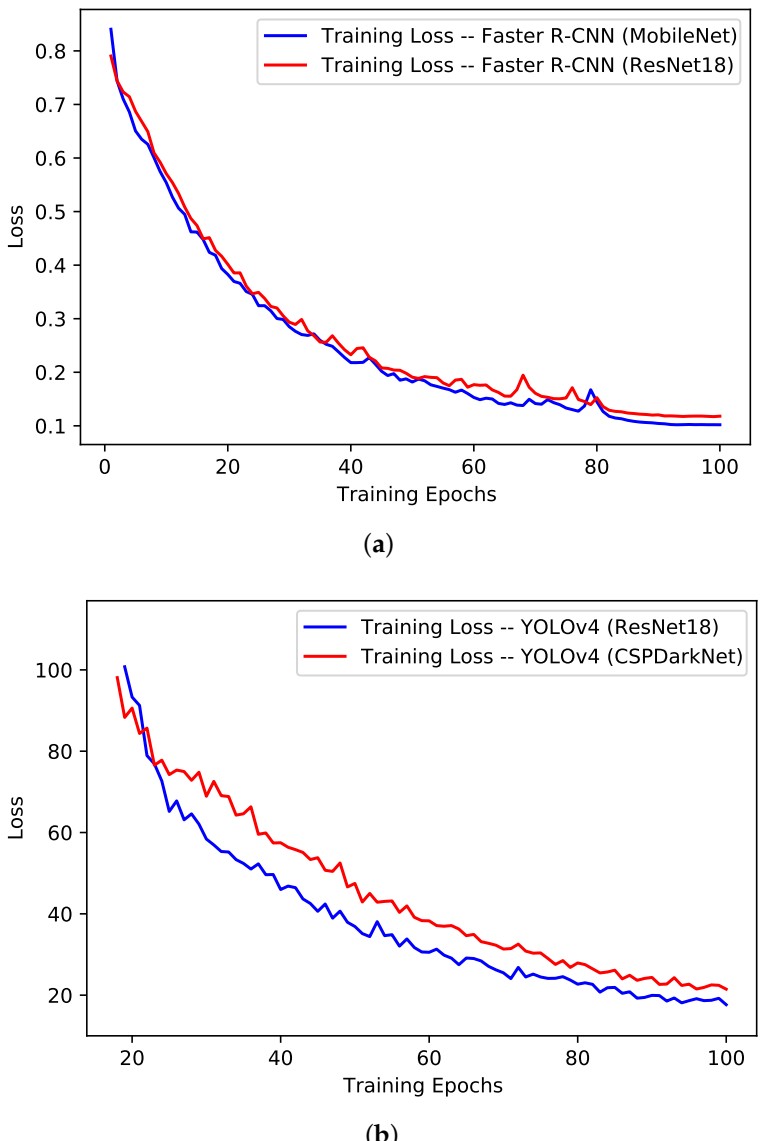

**Figure 4.** Training Loss Curves for the Different Variants of Computer Vision Object Detection Models Implemented for Floodborne Object Type Recognition: (**a**) Faster R-CNN Models, (**b**) YOLOv4 Models.

For the YOLOv4 models (see Figures 4b and 5b), similar to the Faster R-CNN case, both variants performed comparatively similar, with the ResNet18 variant being slightly on the better end in comparison to CSPDarkNet variant. The training loss was found gradually decreasing (i.e., negative exponential) over the training epochs and settled around 20 for the ResNet18 variant, while it was around 24 for the CSPDarkNet variant. The negative exponential trend of training loss is the indication of the normal training process. In terms of validation mAP, curves were stabilized around 76% after the 60th epoch.

The quantitative results from the best validation mAP epoch for each model are presented in Table 2 in terms of training loss, validation mAP, precision and recall. From the results, it can be observed that the Faster R-CNN with the MobileNet backbone model achieved the overall best training results, with a validation mAP of 0.86 at the 90th epoch and a training loss of 0.10. The Faster R-CNN with the ResNet18 backbone model was second best, with a validation mAP of 0.86 and a training loss of 0.35.

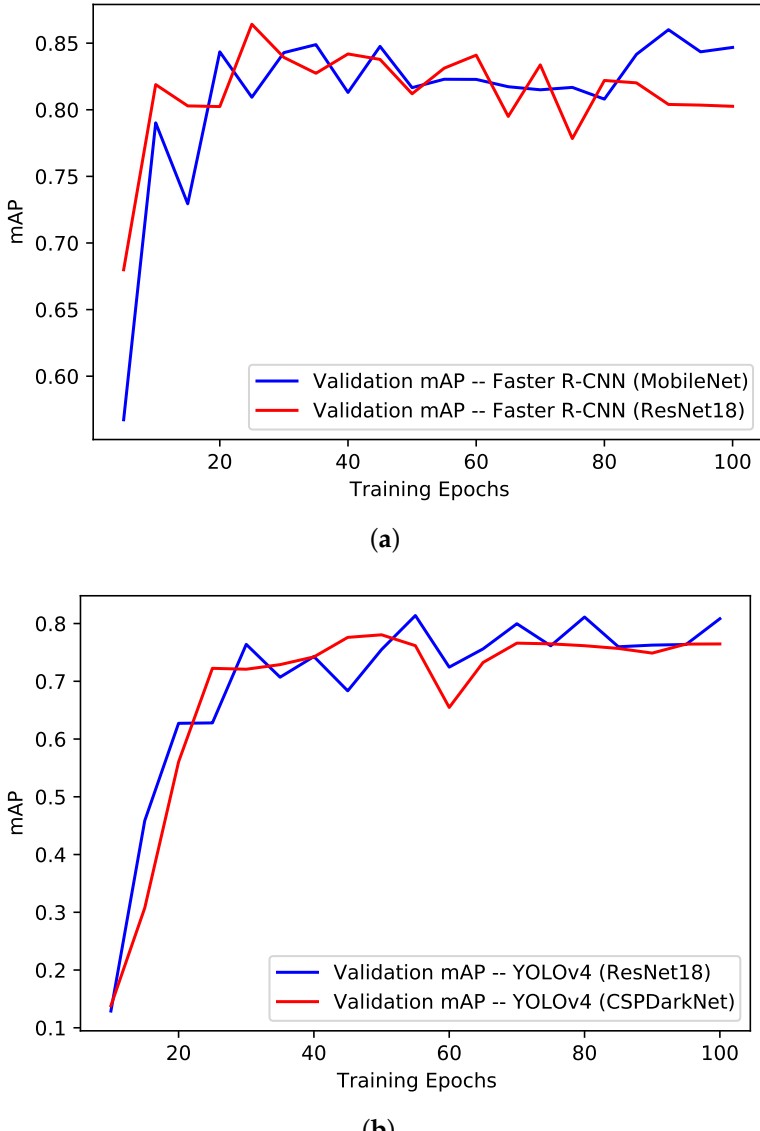

**Figure 5.** Training mAP Curves for the Different Variants of Computer Vision Object Detection Models Implemented for Floodborne Objects Type Recognition: (**a**) Faster R-CNN Models, (**b**) YOLOv4 Models.

**Table 2.** Quantitative Training Results for the Different Variants of Computer Vision Object Detection Models Implemented for Floodborne Objects Type Recognition.

| Model | Training Loss | mAP | Mean Precision | Mean Recall |
|---|---|---|---|---|
| Faster R-CNN Models | | | | |
| MobileNet Backbone | 0.1044 | 0.8601 | 0.2515 | 0.8827 |
| ResNet18 Backbone | 0.3492 | 0.8642 | 0.0713 | 0.8990 |
| YOLOv4 Models | | | | |
| ResNet18 Backbone | 34.87 | 0.8138 | NA | NA |
| CSPDarkNet Backbone | 47.48 | 0.7804 | NA | NA |

Finally, the models' training speeds were also compared by monitoring the training time in seconds per epoch. The training times per epoch for each trained model are presented in Figure 6. It can be observed that the Faster R-CNN model with the MobileNet model took the least time to train (i.e., 30 s per epoch), while the YOLOv4 model with the ResNet18 model took the most time to train (i.e., 160 s per epoch). The longer training times

for the YOLOv4 model variants may be associated with the higher level of complexity of the model and greater number of trainable parameters.

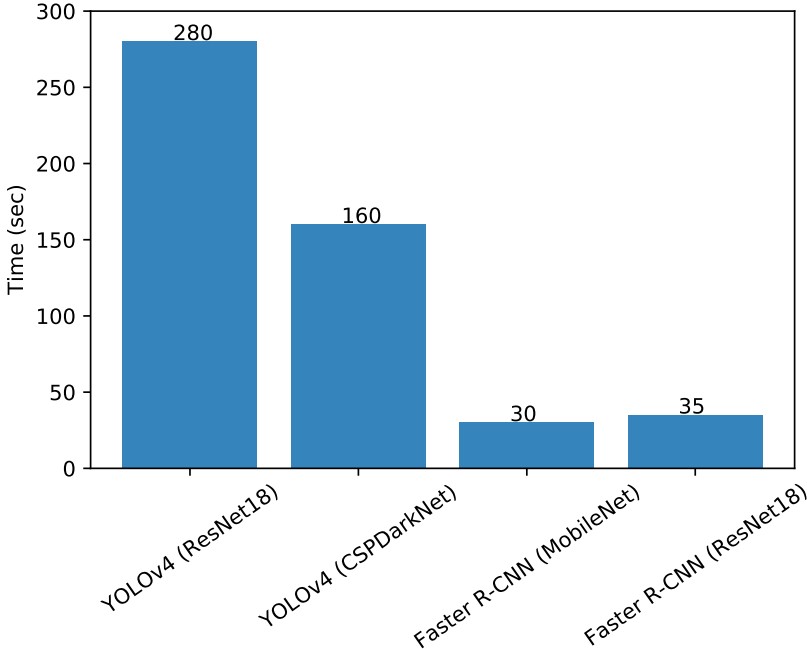

**Figure 6.** Training Time Per Epoch for each Implemented Computer Vision Object Detection Model.

*5.2. Testing Performance*

The trained models were evaluated over the unseen samples from the dataset to assess their test performance. Table 3 presents the qualitative test results for the implemented computer vision model to recognize floodborne objects type in terms of the mAP and AP for each class. From the results, it can be observed that the Faster R-CNN model with MobileNet backbone was able to achieve the best test performance with an mAP of 0.8445, an $AP_{vegetated}$ of 0.7544 and an $AP_{urban}$ of 0.9345. YOLOv4 models were outperformed by a difference of at least 8% in the mAP. The $AP_{vegetated}$ was observed to be consistently lower in comparison to the $AP_{urban}$. This may be attributed to the more vegetated annotations in the challenging real dataset samples in comparison to the lab-simulated controlled samples. The performance of the computer vision detection models was found comparable to the reported performance by Lin et al. [57] (i.e., mAP of 70%), Cheng et al. [46] (i.e., mAP of 43%) and Liseshout et al. [48] (i.e., mAP of 68.7%).

**Table 3.** Quantitative Testing Results for the Different Variants of Computer Vision Object Detection Models Implemented for Floodborne Object Type Recognition.

| Model | mAP | $AP_{vegetated}$ | $AP_{urban}$ |
|---|---|---|---|
| Faster R-CNN (Resnet18 Backbone) | 0.8007 | 0.7236 | 0.8778 |
| Faster R-CNN (MobileNet Backbone) | 0.8445 | 0.7544 | 0.9345 |
| YOLOv4 (ResNet18 Backbone) | 0.7826 | 0.7393 | 0.8331 |
| YOLOv4 (CSPDarkNet Backbone) | 0.7616 | 0.7115 | 0.8114 |

## 6. Discussion

The experimental investigations reported in Section 6 indicated that the computer vision models have significant potential for the automated recognition of floodborne object types in images. The best performing model (i.e., Faster R-CNN with MobileNet backbone) was able to achieve the mAP of 0.84 on a relatively challenging dataset. Figures 7 and 8 show a few samples from the results where the best-performing computer vision model predicted floodborne object types accurately and made mistakes, respectively. From the

samples, it can be observed that the model made a mistake while detecting vegetation. In one instance, the model mistakenly detected a background tree as vegetation, while in another instance, it mistakenly detected a grassy region as vegetation. These false predictions may be associated with the similarity in the visual appearance of vegetation with the natural backgrounds containing a lot of trees and grass. In a third instance, the model missed a few detections and partially detected the vegetation in the scene. This may be attributed to the clutteredness of the scene and the presence of connected floodborne objects. The results for the lab-simulated controlled samples were observed to be very high, mainly because of the presence of no background noise and clear differentiation between different floodborne object types.

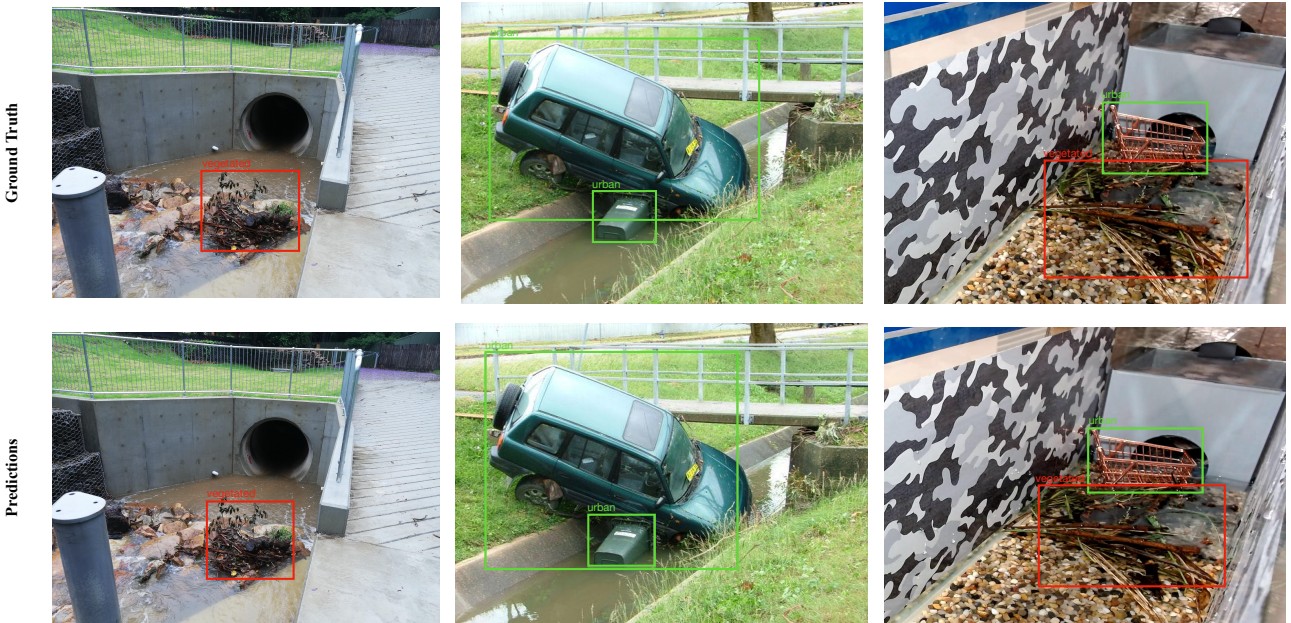

**Figure 7.** Sample of Correct Predictions by the Faster R-CNN with MobileNet Backbone Model.

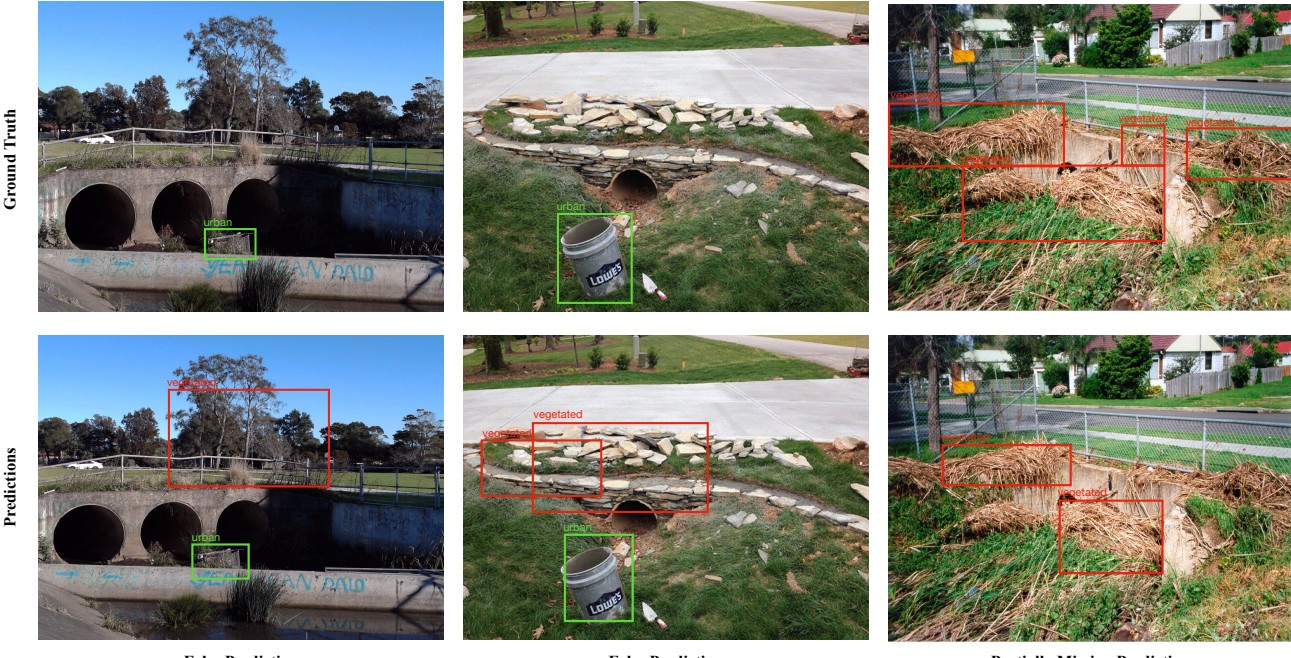

**Figure 8.** Sample of False Predictions by the Faster R-CNN with MobileNet Backbone Model.

The presented research is in line with the literature corresponding to the quantification and budgeting of vegetation in rivers towards determining the induced fluxes. The benchmark studies in this regard [54,55] implemented the classical image processing techniques and cannot be considered as the generalized solution to the problem. The classical solution may work specifically for the quantification of the vegetation in a specific use-case; however, it is likely to fail in varying lighting conditions (adverse weather) and composite types of floodborne objects. On the other hand, it has been established from the literature that deep-learning-based approaches are way more efficient and offer robust generalized performance. In this context, the presented research in this article is the first step towards using automated computer vision analysis of floodborne objects for incorporating the information into flood mitigation and management policies/systems. The resulting information about the types of floodborne objects along with other known information can potentially be used for issuing early warnings and/or evacuations. For example, given the information about the hydraulic capacity of a structure, the upstream discharge through sensors and rainfall information from radar along with the types and sizes of floodborne objects, the system can be configured to estimate the flooding outcomes, damage to structure and likelihood of structure failure.

## 6.1. Research Implications

From the utility perspective, such a system is proposed to be deployed either pointing towards the hydraulic structures to detect the accumulated floodborne objects for maintenance purposes or pointing away from the structure to detect the incoming floodborne objects for blockage-related flood mitigation. The real-world images used in this research for training were not captured taking into account the utility of such a system; rather, they are randomly captured by flood management officials, and therefore, they contain a lot of background noise visually similar to vegetation, degrading the detection performance. However, these problems can be avoided by calibrating the camera in such a way that it captures only the region of interest. For the first use case, the camera should be pointed right at the hydraulic structure, avoiding any background, which will make it easier for the model to detect the correct floodborne objects accumulated at the structure. The information extracted by the system will help in better understanding the floodborne objects' accumulation patterns, in monitoring different ways the floodborne objects interact with structures and in making maintenance solutions. For the second use case, the camera should be pointing away from the structure (most probably a bridge on a river) such that it only captures the water background and not the background vegetation. Vegetation or urban objects floating in the water will be easier for the model to detect because of the clear visual differentiation between water and floodborne objects. In addition to the floodborne objects type detection from the camera system, a LIDAR sensor may be used to determine the volume of incoming floodborne objects, both of which can then be incorporated into the flood models towards mitigating the floods. The information extracted by the system will be incorporated in hybrid data-driven self-correcting models towards better and adaptive flood modelling. From a practical perspective, the models trained in this research can serve as baseline models for the pilot installation and will be fine-tuned over time more data become available to achieve the optimized performance. Graphical illustrations of culvert and river use cases are presented in Figure 9a,b, respectively.

## 6.2. Limitations and Future Directions

The availability of the relevant visual floodborne objects data from real flooding events is one of the key challenges of the presented research. One potential approach to address the data availability problem is to use the synthetic images generated from different simulated platforms, including NVIDIA Isaac replicator, FLOW 3D, Generative Adversarial Networks (GANs) and style transformations. In addition, as an initiative, such a pilot system installed on a real site will be able to generate plenty of relevant data during the first year duration. Although the approach of using the simulated data towards enhancing the

dataset is reported useful in most of cases, to ensure its usefulness, a detailed investigation on the impacts of different types of simulated data would be needed. It is anticipated that the performance of computer vision models will degrade with the use of simulated data; therefore, an investigation of determining different data mixes with different ratios needs to be performed. The system proposed in this research is only capable of recognizing if a given floodborne object is vegetation or an urban object. However, in the future, more detailed analyses are planned, including detecting different types of vegetation and urban objects at a subclass level and detection of the composite floodborne objects where both vegetation and urban objects are present. The quantification of the detected floodborne objects is the ultimate goal to bring the presented research in line with the literature. It is planned to make use of a camera matrix and proper calibration to estimate the size of the detected floodborne objects in pixels which can then be translated to volume.

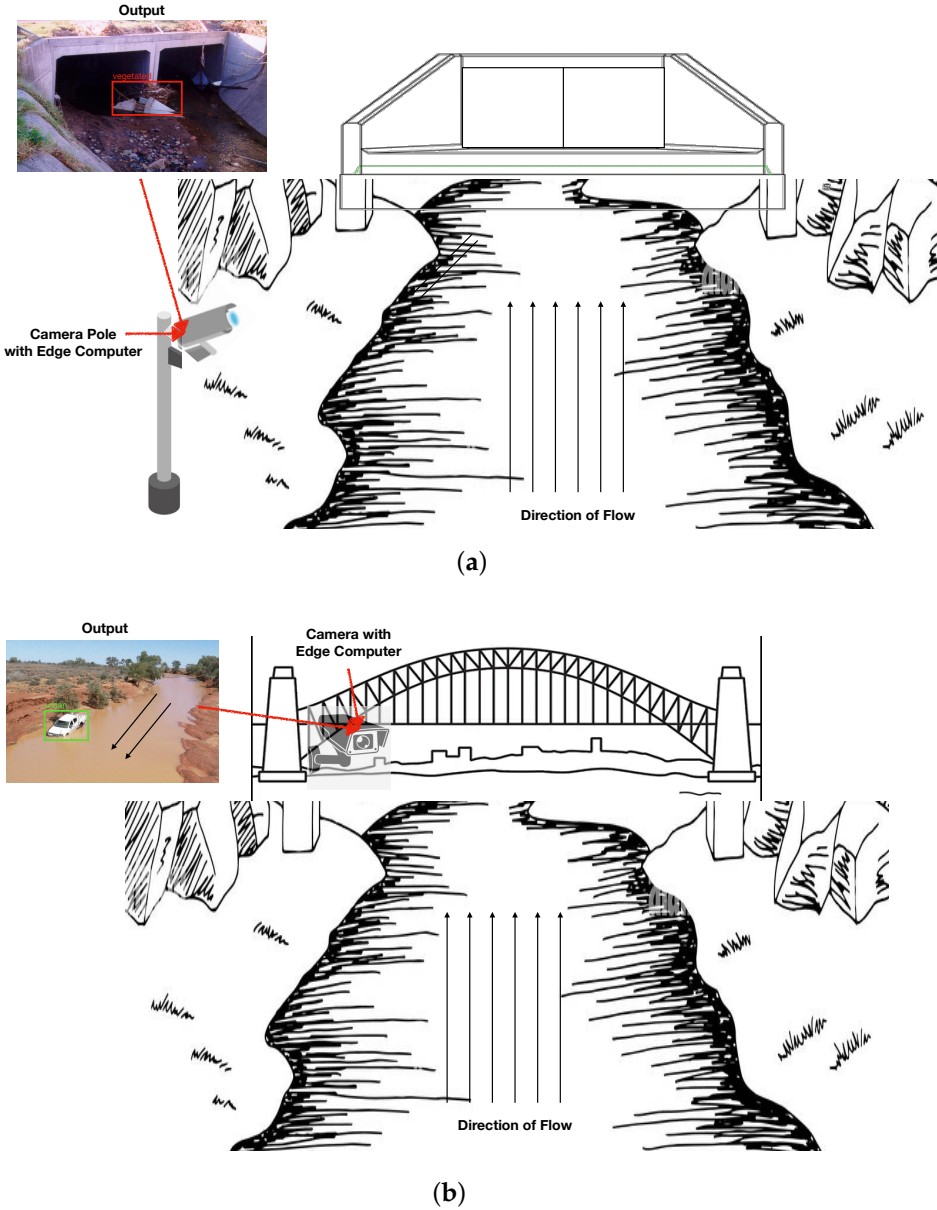

**Figure 9.** Graphical Illustration of Two Potential Use Cases for the Proposed Floodborne Object Type Recognition using Computer Vision: (**a**) Culvert Use Case, (**b**) River Use Case.

## 7. Conclusions

Computer vision object detection models have been successfully implemented for the recognition of floodborne object types. Multiple variants of the Faster R-CNN and YOLOv4 models were trained using the FORD dataset and were evaluated in the context of training and testing performance. From the experimental investigations, the Faster R-CNN model with MobileNet backbone was able to achieve the best mAP of 84%. In terms of class-wise AP, the best model achieved 75% for vegetation and 93% for urban objects. Two potential use cases (i.e., culvert use case, river use case) were also described to demonstrate the practical usability of the proposed research. The availability of relevant data and the challenging nature of vegetation detection were highlighted as potential hindrances that need to be addressed in the future. Deployment of pilot projects to collect data and the use of simulated data generation platforms including GANs, NVIDIA ISAAC replicators and style transfers are potential future research directions. In addition, investigating the impact of simulated data on the performance of computer vision models is also among the tasks planned to be performed in the near future.

**Author Contributions:** Conceptualization, U.I., M.Z.B.R., J.B., N.H. and P.P.; methodology, U.I. and J.B.; software, U.I. and J.B.; validation, U.I., J.B., N.H. and P.P.; formal analysis, U.I.; investigation, U.I.; resources, U.I., M.Z.B.R., J.B. and P.P.; data curation, U.I.; writing—original draft preparation, U.I.; writing—review and editing, U.I., M.Z.B.R., J.B., N.H. and P.P.; supervision, J.B. and P.P. All authors have read and agreed to the published version of the manuscript.

**Funding:** This work was supported by the Wollongong City Council (WCC) in partnership with Shell-harbour, Kiama and Shoalhaven Councils, Lendlease and the University of Wollongong's SMART Infrastructure Facility. This program also received funding from the Australian Government under the Smart Cities and Suburbs Program (SCS69244). We also gratefully acknowledge the support of NVIDIA Corporation with the donation of the Titan RTX GPU used for this research.

**Institutional Review Board Statement:** Not applicable.

**Informed Consent Statement:** Not applicable.

**Data Availability Statement:** Not applicable.

**Acknowledgments:** I would like to thank the WCC for providing resources and support to carry out this study.

**Conflicts of Interest:** The authors declare no conflict of interest.

## Abbreviations

The following abbreviations are used in this manuscript:

| | |
|---|---|
| YOLO | You Only Look Once |
| CNN | Convolutional Neural Network |
| RPN | Region Proposal Network |
| FORD | Floodborne Objects Recognition Dataset |
| AP | Average Precision |
| mAP | Mean Average Precision |
| IoU | Intersection of Union |
| SVM | Support Vector Machine |
| MAP | Feature Map Attention |
| FLS | Forward Looking Sonar Images |
| WCC | Wollongong City Council |
| ICOB | Images of Culvert Opening and Blockage |
| VHD | Visual Hydraulics-Lab Dataset |
| CSP | Cross Stage Partial Connections |
| SAT | Self Adversarial Training |
| WRC | Weighted Residual Connections |
| CmBN | Cross Mini-Batch Normalizations |

| TAO | Train Adapt Optimize |
|---|---|
| SGD | Stochastic Gradient Descent |
| Adam | Adaptive Momentum |
| GPU | Graphical Processing Unit |
| GANs | Generative Adversarial Networks |

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
