# Peer review of "Floodborne Objects Type Recognition Using Computer Vision to Mitigate Blockage Originated Floods"

_water, doi:10.3390/w14172605_

Round 1

Reviewer 1 Report

The article submitted for review is interesting. The topic of the work is novel, the structure and layout are correct. The work is written neatly and fluently. The work fits to the scope of Special Issue and maybe published after same editorial corrections.

I have one suggestion. The authors could, however, explain more broadly why they distinguish the origin of the debris. What is it for? What's the benefit?

Author Response

Authors thank the reviewer for providing the valuable feedback on the manuscript. The response to comments has been provided in the attached file.  

Reviewer 2 Report

Summarized of Article:

The study aims to use computer vision technologies (vision sensors) for automated element identification accumulated at a hydraulic structure during a flood. They compare field data with laboratory experiment simulation. The authors demonstrate the positive potential of the technology.

General comments and my recommendation as reviewer:

While the topic is of great interest, in my view the manuscript needs major changes and a full revision before it could become a significant contribution. I am not convinced that the manuscript is ready to be accepted in its current shape. The manuscript needs some structural revision suggested in the detailed comments. As the results are not adequately discussed in the light of the most important papers on the topic. The results need to be discussed in depth, as several papers have previously tackled the topic (e.g., McVicar et al., 2009. https://doi.org/10.1002/esp.3240; Ghaffarian et al., 2021. https://doi.org/10.5194/esurf-2020-96).

Detailed comments:

Abstract

The term debris is no longer used when referring to vegetation, in particular to wood.

Thus, I would like to suggest removing the term “debris” throughout the entire manuscript, at least when it refers to vegetation and wood. As reported in the preface of the book of the First International Conference of Wood in World Rivers (Gregory, S. V., K. L. Boyer, and A. M. Gurnell, editors. 2003. The ecology and management of wood in world rivers. American Fisheries Society, Symposium 37, Bethesda, Maryland), the term “debris” was first used to refer to the wood slash and debris left on the land and in the stream after timber harvest. For this reason, the term negatively connotes garbage or trash to the general public. The debate was also reported during the Third and Fourth International conference of Wood in World Rivers where the audience positively accepted to discourage further the use of the term “debris”, encouraging the only use of the word “wood”.

Line 1-5: There are repetitions. Generate one sentence.

Line 7: what do you mean with “movement”? The kinetics?

Line 7-10: Please rephrase. You could define them as “vegetation (i.e., logs, branches, shrubs, entangled grass)” and “urban debris (i.e., vehicles, bins, shopping carts, and building waste materials)”.

Introduction

Line 65-66: Authors talk about Faster R-CNN and YOLOv4 models but they did not explain what they are. They do it in subchapters 4.1 and 4.2. The models should be introduced before (even briefly) in order to facilitate understanding for a general reader. Thus, I suggest merging chapter 4 with the introduction.

Line 54-56: It’s not totally true. I) The porosity depends on the type of vegetation. A tree with rootwads is more porous than a tree without rootwads (Ravazzolo et al., 2022. https://doi.org/10.1029/2021WR031403). Ii) The diameter strongly influences the depth of flow required to entrain and transport logs (Bilby & Ward, 1991. https://doi.org/10.1577/1548-8659(1989)118<0368:CICAFO>2.3.CO;2; Braudrick & Grant, 2000. https://doi.org/10.1029/1999WR900290). Increasing the log density also increases the water depth required for entrainment (Braudrick & Grant, 2000). Iii) the hydraulic blockage also depends on the transport regime (Braudrick et al., 1997. https://doi.org/10.1002/(SICI)1096-9837(199707)22:7<669::AID-ESP740>3.0.CO;2-L; Ruiz-Villanueva et al., 2019. https://doi.org/10.1002/esp.4603; Schalko et al., 2019. https://doi.org/10.1080/00221686.2019.1625820).

Line 54-58: Summarizing, authors should demonstrate, in a stronger way, the differences between vegetation and urban debris. They should also add references supporting their statements.

 Authors should demonstrate the importance of the work conducted.

Related work

Line 83-88 can be removed.

Also Benacchio et al., 2017. http://dx.doi.org/10.1016/j.geomorph.2016.07.019; McVicar et al., 2009. https://doi.org/10.1002/esp.3240

Floodborne Debris Recognition dataset (FDRD)

The dataset should be moved to the methods chapter

Background to Computer Vision Object Detection Models

Please, read my comment about Line 65-66

Line 214: what does “loss” be? “training loss” is also introduced in line 248. It should be explained given that it is important for understanding your results.

Methodology and Experimental Protocols

The chapter needs a better description of the five stages. In each stage, the authors should explain in detail the process used. At this stage, the description is too general and difficult to understand.

For example:

Line 240: What do you refer with “the label”? what is KITTI format?

Line 240-241: the sentence should be better explained.

Line 242: with “data” you mean “image”, right? So, call it "image" to be clearer.

Line 244-246: this distinction should be clarified before, when you introduce the two models.

Line 247: NVIDIA TAO toolkit with fundamental data augmentation. What do you mean?

Line 249: What do you mean with mAP? (Then, in line 311: what do you mean with AP?)

The figure can be improved by adding the number of the stages. In each box of the diagram, you could add a title like: “Stage I: Image processing”; “Stage II: Model selection” (it is only a suggestion).

Experimental Protocols and Evaluation Measures

This subchapter should be merged with chapter 5. You are introducing NVIDIA TAO in line 255 when you already talked about it in line 247.

Discussion of the Results

The title can be changed to “Discussion”

Authors should relate their results with the literature. The discussion should show how your findings fit with existing knowledge, what new insights they contribute, and what consequences they have for theory or practice. Do your results agree with previous research? Are your findings very different from other studies? Do the results support or challenge existing theories? Are there any practical implications? Your overall aim is to show the reader exactly what your research has contributed and why they should care.

The chapter might be divided into subchapters. For example, a sub-chapter talking about limitations, another about implications and perspectives for future studies.

Figures: Indicate flow direction; What do the red and green boxes indicate?

Author Response

(The authors gave the same response as above.)

Reviewer 3 Report

Please see the attached word file.

Author Response

(The authors gave the same response as above.)

Round 2

Reviewer 2 Report

The manuscript has been improved by the authors and is now clearer in the methods, results and discussion; however, they did not address my concern about the use of the word “debris”. As I suggested in the previous revision of the manuscript, I would encourage the authors to avoid the use of “debris” when refer to “wood” or “vegetation”. Although you agree with me in modifying the text, I am still reading “debris” in several parts of the manuscript (line 483, 486). Also, you often used the word "debris" to refer to both urban and vegetation elements, when vegetation we have agreed not to be (e.g., lines 419, 455, 467, 488). Please, modify the text using the word “elements” or “objects”.

Minor edits:

Line 308: remove the parenthesis.

Figures 7 and 8: please add the label “urban” and “vegetation” in all the boxes.

Author Response

Authors appreciate the reviewer for providing the valuable feedback in the second round of review. The manuscript has been revised and response to each comment has been provided in the attached file.
